# Spleen: Reparative Regeneration and Influence on Liver

**DOI:** 10.3390/life12050626

**Published:** 2022-04-22

**Authors:** Andrey Elchaninov, Polina Vishnyakova, Gennady Sukhikh, Timur Fatkhudinov

**Affiliations:** 1Laboratory of Regenerative Medicine, National Medical Research Center for Obstetrics, Gynecology and Perinatology Named after Academician V.I. Kulakov of Ministry of Healthcare of Russian Federation, 117997 Moscow, Russia; vpa2002@mail.ru (P.V.); gt_sukhikh@bk.ru (G.S.); 2Histology Department, Medical Institute, Peoples’ Friendship University of Russia, 117198 Moscow, Russia; tfat@yandex.ru; 3Laboratory of Growth and Development, Scientific Research Institute of Human Morphology, 117418 Moscow, Russia

**Keywords:** spleen, regeneration, resection, transplantation, liver

## Abstract

This review considers experimental findings on splenic repair, obtained in two types of small animal (mouse, rat, and rabbit) models: splenic resections and autologous transplantations of splenic tissue. Resection experiments indicate that the spleen is able to regenerate, though not necessarily to the initial volume. The recovery lasts one month and preserves the architecture, albeit with an increase in the relative volume of lymphoid follicles. The renovated tissues, however, exhibit skewed functional profiles; notably, the decreased production of antibodies and the low cytotoxic activity of T cells, consistent with the decline of T-dependent zones and prolonged reduction in T cell numbers. Species–specific differences are evident as well, with the post-repair organ mass deficiency most pronounced in rabbit models. Autotransplantations of splenic material are of particular clinical interest, as the procedure can possibly mitigate the development of post-splenectomy syndrome. Under these conditions, regeneration lasts 1–2 months, depending on the species. The transplants effectively destroy senescent erythrocytes, assist in microbial clearance, and produce antibodies, thus averting sepsis and bacterial pneumonia. Meanwhile, cellular sources of splenic recovery in such models remain obscure, as well as the time required for T and B cell number reconstitution.

## 1. Introduction

The spleen is the largest lymphoid unpaired parenchymal organ of the abdominal cavity found in all vertebrates. The spleen is functionally and morphologically divided into the red and white pulp with a marginal (in rodents) or perifollicular zone (in humans) between them (Figure 1) [1]. Blood circulation in the spleen is open: blood enters the tissues via the trabecular artery and becomes the central artery which gives rise to many branches which enter the red pulp and surround the white pulp [2]. The red pulp is responsible for blood filtration and removes opsonized, damaged, and dying cells from circulation. It also serves as the depot of the organism’s iron, as well as red blood cells, monocytes, and platelets. The spleen, as the largest secondary lymphoid organ, contains about a quarter of the body’s lymphocytes and initiates an immune response to blood antigens [3]. Although adaptive immune responses to such antigens are realized in the white pulp, the cells of innate immunity (neutrophils, monocytes, dendritic cells, and macrophages) could be easily found as residents of red pulp. The white pulp is divided into T- and B-zones, similar to the lymph nodes of the immune system. The T cell zone is also called the periarteriolar lymphoid sheath because it surrounds the central arterioles and is composed of lymphocytes, reticular cells, and reticular fibers [3]. B cell zones represent follicles where clonal expansion of activated B cells can take place [4]. The marginal zone serves to monitor the circulation for antigens and pathogens and plays an important role in antigen processing and for lymphocytes releasing from the circulation and entering the white pulp. The marginal zone mainly contains macrophages and a special subset of B cells (marginal zone B cells) [1].

Total resection of the spleen is an option in a number of clinical situations including traumatic injury, thrombocytopenia, and severe portal hypertension [5]. Despite the apparent simplicity of the procedure, 2–10 years in the aftermath, a number of patients develop complications collectively referred to as post-splenectomy syndrome, most typically manifested by recurrent infections of varying severity [5]. Clinical examples demonstrate a different patient survival rate after splenectomy [6]. In the early work of Bonnet-Gajdos and colleagues, it was shown that splenectomy performed in 21 patients with age from one to 23 years old had no pronounced negative clinical manifestations and did not affect the normal maturation of children [7]. In another study with a 19/21 survival rate, splenectomy caused by sporadic haemophagocytic lymphohistiocytosis (HLHs) showed a real diagnostic benefit in establishing the cause of HLHs and had a therapeutic effect [8]. Splenectomy leads to an increased risk of septic complications associated with high mortality, the most serious of which is the development of functional hyposplenism associated with overwhelming post-splenectomy infection (OPSI), which can progress from flu-like illness to fulminant sepsis in a short period of time and is accompanied by high mortality [9]. The first descriptions of OPSI date back to 1952, when King and Schumaker first described bacterial sepsis after splenectomy in infants and children [10]. In a large retrospective study of 6942 patients, Bisharat and colleagues showed a low risk of postsplenectomy sepsis; however, approximately 40–50% of all people who develop postsplenectomy sepsis died [11]. Now it is known that the predominant causative agents of postsplenectomy sepsis are resistant to phagocytosis encapsulated bacteria, Streptococcus pneumoniae [12]; therefore, the long-term antibiotic therapy is designed to prevent the development of OPSI after splenectomy [13]. Some studies also claim increased risks of tumorigenesis after splenectomy [6,14,15,16]. Such complications indicate significant problems in both humoral and cellular wings of immunity. Considering the prevalence of such complications, regeneration of the spleen is extremely relevant. Apart from the issues of splenic repair per se, this review considers effects of the spleen on hepatic recovery after various kinds of damage. These effects are substantial due to the close anatomical and functional relationship between the two organs.

## 2. Splenic Regeneration after Resections

The most common indications for splenectomy are a wide variety of diseases and conditions: raptured or enlarged spleen, blood disorders like idiopathic thrombocytopenic purpura and thalassemia, cancer (chronic lymphocytic leukemia, Hodgkin’s lymphoma, and non-Hodgkin’s lymphoma), infection, etc. The study of spleen regeneration after splenectomy is often limited to patients and small laboratory animals, while for larger species, splenectomy has only been studied in the context of survival. Thus, it was shown that post-splenectomy, post-operative survival rate was 52% for dogs [17], 87.5% for Theileria haneyi-infected, splenectomized horses [18], and almost complete post-operative recovery was observed for cattle in more earlier works [19,20].

The first studies on mammalian spleen regeneration date back to the 19th century [21,22]. Dedicated research on the dynamics, cellular sources, and spleen microanatomy after resections peaked in the 20th century [21,22]. Over recent years, the focus has shifted from post-splenectomy regrowth to post-transplantation recovery. Data on spleen regeneration after resection are summarized in Table 1.

The earliest known experimental studies on regeneration of the spleen after resections were carried out in the 19th century by Jean-Marie Philipeaux [21]. In early experiments, resections of the spleen ended with restoration of its shape and mass; occasionally, the remnant overgrew the initial size of the organ. Later on, these results were questioned and, by the turn of the century, the general opinion was that the spleen does not regenerate after resections and the site heals by scarring [21].

A bunch of experimental studies on regeneration of the spleen after resections was implemented by academic staff of the Laboratory of Growth and Development at the Scientific Research Institute of Human Morphology, Moscow, Russia. One series involved mice undergoing 50% splenectomy. The extent of regenerative outcomes considerably varied: after 1 month, most of the remnants grew significantly in length and volume, while preserving the architecture and showing no outgrowth at the wound surface. It should be noted that, even in cases of intensive regrowth, the organs never reached their initial size. Moreover, in a number of cases, the remnants progressively shrunk [22,23,24].

Resections of 90% of the splenic volume in mice produced functional spleen equivalents of loosely cubic shape in 38 days [21,22]. Regeneration was accompanied by an increase in the density of lymphoid follicles and their relative area, as assessed by light microscopy [25,26]. The earliest observable reaction of splenic tissues to the resection involved the emergence of macrophages burdened with particles of destroyed nuclei. Six hours post resection (p/r), reticular tissue cells started to proliferate and were followed by erythrocyte and lymphocyte lineages enriched in cells with blast morphology [21,22]. Erythrocyte progenitors reached their maximal mitotic activity by day 3 p/r. Interestingly, repeated resections promoted a sharp increase in the content of erythrocyte lineages within the remnant and a reciprocal decrease in the content of lymphocyte lineages [21,22]. Transplantations of cells isolated from intact and regenerating spleens to lethally irradiated mice yielded equal numbers of colonies, indicating the rapid recovery of stem cell populations [27]. The rates of recovery for T and B cell populations in regenerating spleens differed: the slower proliferation of T cells resulted in their significantly reduced numbers as late as 3 months after the resection [21,22].

In addition to the morphological and cell lineage studies, the focus was expanded to the functional profiles of splenic lymphocytes, including the production of antibodies and participation in cellular immunity reactions. Following the resection and regrowth, splenic lymphocytes produced lower amounts of antibodies and showed weaker graft-versus-host reactions compared with the cells from intact spleens [28]. The observed functional decline was explained by the enrichment with immature, functionally compromised lymphocytes in the course of regeneration [21,22].

Species–specific differences in splenic regeneration are pronounced. Rabbits, by contrast with mice and rats, are incapable of splenic regrowth [25]. The difference is interesting in terms of adaptive immunology: rabbits respond to splenectomy by compensatory increase in the volume of lymph nodes, which acquire structural features of the spleen. Splenic regeneration in rabbits can be induced chemically [29]. In other model animals, regeneration rates significantly depend on age (higher in younger animals) and season (higher in summer). In addition, the regrowth can be inhibited by thymectomy [28] and stimulated by injections of certain antigens [21,22].

Together, these studies indicate that classical experiments on spleen resection unequivocally indicate its ability to regenerate, which is accompanied by the preservation of tissue architectonics and an increase in the size of lymphoid follicles. In the next section, we will turn to the analysis of heterotopic (not in original location) transplantations of the spleen as a frequent model for studying regeneration.

**Table 1 life-12-00626-t001:** Spleen regeneration after resection.

	Authors	Volume of Resection	Regeneration Period	Antibody Production	T Cell Activity	Other
1.	L. D. Liozner and Kharlova [24]	50%	30 days	Decreased production of antibodies	Decreased T cell activity	The observed functional decline was explained by the enrichment with immature, functionally compromised lymphocytes in the course of regeneration
2.	Cameron and Rhee [23]	50%	30 days			
3.	Macka and Scott Polland [25]	50%	30 days			An increase in the density of lymphoid follicles and their relative area
4.	Kharlova [22]	90%	38 days			Slower recovery of T cells compared to B lymphocytes
5.	Pouché et al. [26]	50%	90 days			The results of histologic study demonstrate a readjustment of the vascular net and the lymphoid tissue of the white pulp

## 3. Autotransplantations of the Spleen

The orthotopic regeneration of the spleen is clinically unfeasible. At the same time, unattended splenectomies are fraught with delayed immunological complications [5]. The problem can be solved by transplantation of splenic material to heterotopic locations.

The idea was born in connection with clinical cases of spontaneous splenic engraftment. In the aftermath of splenectomy, functionally active fragments of splenic lymphoid tissue may occasionally settle in the abdominal cavity, resembling accessory spleens. This phenomenon is known as ‘splenosis’ [30,31].

Studies on the heterotopic recovery of autologous splenic grafts in rodent models began in the early 20th century. Two major sites of heterotopic engraftment were abdominal (mostly intraomental) and subcutaneous. The course of regeneration for these two sites was similar and took about 1 month [32,33]. Data on heterotopic spleen transplantation are summarized in Table 2.

An early series of such experiments was, oddly enough, performed on rabbits [34,35], incapable of orthotopic splenic regrowth under standard conditions [25]. Nevertheless, the splenic architecture was restored within 80 days after subcutaneous autotransplantations. In younger rabbits or rabbits after total splenectomy, the regeneration proceeded faster [34,35].

These pioneering models were meticulously reproduced in other species of laboratory animals [36]. In rats and mice, the abdominal engraftments proceeded similarly and lasted about 1 month [33]. Other studies indicated longer regeneration in rats, about 2 months [23,37]. The early stage (days 0–3 post-transplantation, p/t) was marked with degradation of the tissue architecture and massive death of lymphocytes. At 12–18 h p/t, two zones could be clearly distinguished within the graft: the central zone of necrosis and the peripheral zone of viable cells, with reticular cells projecting towards the central zone. First capillaries emerged at 48 h p/t. On day 7–9, clusters of cells with round dark nuclei and lymphocytic morphologies appeared in the peripheral zone and around capillaries. Starting from day 13–15, formation of lymphoid follicles with enlarged lymphocyte-like cells was evident. By day 29 p/t, the graft acquired characteristic tissue architecture of the spleen [23,33,38,39] and comprised differentiated B cells and red pulp macrophages [40].

Consistent with the previous findings on rabbits, in rat models, the engraftment was more robust in younger animals, with accelerated vascularization and re-innervation of the grafts, and higher numbers of Ki67+ cells and their distributions typical of the intact spleens in young rats [39,41,42]. In contrast to rabbit models, in rats, the engraftment was more robust after incomplete splenectomies [23,33,43].

Over recent decades, these techniques were refined. The optimal site for peritoneal engraftments is the greater omentum, although autologous transplantations to the mesentery or the inside of abdominal wall work as well [44,45,46]. Moreover, the whole ectomized spleen can be attached surgically to the liver, with successful outcomes [47].

Subcutaneous splenic grafts provide a major alternative, although the scenario is basically similar to the intraperitoneal splenic regeneration. In rats, the subcutaneous engraftments are finalized within 1–1.5 months p/t. Identical to the peritoneal grafts, the early phase is marked with necrosis. Subsequent morphological landmarks are blood vessel ingrowth (day 3) and accumulation of larger lymphocytes with dark nuclei in peripheral parts of the transplant (day 7). By day 14, the lymphocytes increase in number. By day 28, the transplant recovers characteristic splenic architecture with red and white pulp compartments [32]. These observations are corroborated by more recent studies of neovasculogenesis in murine transplants, with precise measurements enabled by injection of tracers (fluorescent polystyrene microspheres) and electron microscopy. Although the new vessels at the periphery of the grafts started to form on day 3 p/t, microcirculation within the marginal zone of white pulp remained rudimentary until week 10 p/t [48].

Most of the experiments indicate the universal capacity of autologous splenic grafts to regenerate at a variety of anatomical sites. At the same time, several studies argue that subcutaneous environments are less favorable due to the lower rates of angiogenesis and specify the greater omentum as the optimal site for the engraftment [49]. Such conclusions are based on functional metrics: only intraomental transplants, as opposed to intramuscular, etc., ensured the ‘intact’ rates of pneumococcal clearance after regeneration [50].

Comprehensive functional assessment of the outcomes is important in connection with the clinical relevance of splenic autotransplantations. In this regard, hematopoietic status and resistance to particular infections and tumors in the aftermath of the intervention are chief indicators of its success or otherwise.

Within 2 weeks p/t, the autografts begin to effectively sort erythrocytes, with the concomitant clearance of senescent erythrocytes from the blood [51,52,53]. The first week p/t is marked with a sharp increase in platelet counts and fibrinogen levels. These indicators return to normal within 3–6 weeks as the splenic tissue regenerates heterotopically [51,52].

The main clinical goal of splenic autografts is to prevent the development of severe infectious complications [54,55,56]. According to experimental findings, the prospects are realistic. For instance, splenic autotransplantations to the great omentum in rat models ensured the anti-pneumococcal defense [57,58,59], and similar effects were achieved with subcutaneous grafts [60]. Primary indicators are the ability of splenic autografts to produce antibodies and promote bacterial clearance [49]. After experimental autotransplantations in laboratory animals, total blood levels of IgM are restored within 8 months [61,62]. In functional tests with pneumococcal vaccination, autologous transplantations provide partial response in the form of specific IgM and IgG antibodies, albeit not to all pneumococcal serotypes [40]. Apart from the antibody production, the grafts effectively promote bacterial clearance; this was demonstrated for infections with *E. coli*, as well as pneumococci [38,50,63]. Participation of splenic autografts in the immune response to *Staphylococcus aureus* was explored in mouse model. Splenectomized animals with splenic autografts produced lesser colonies in seeded blood cultures along with higher titers of *S. aureus*-specific IgM and IgG1 in comparison with flat splenectomies [64]. Despite the encouraging results, the influence of autologous grafts on the circulating pools of T and B cells remains understudied. In several studies, splenic autografts did not rescue the lymphocyte blood counts, and after 8 months the circulating numbers of both CD3+ T and CD19+ B cells remained reduced [52,61,62]. In other studies, the ability of spleen autografts to confer immunity against fatal infections was either negligible, or the effect was short-term [49].

In translational perspective, it is important to determine the critical mass of splenic transplant, competent of protecting the body from fatal infections. Although such a value may be of pure theoretical interest [49], several studies loosely define it as 30–50% of the intact spleen mass [49,63].

Efficacy and safety of autologous splenic grafts were assessed in a number of clinical trials (≥18). Sixteen of those involved engraftments in the greater omentum and two trials involved engraftments at retro- and extraperitoneal locations. The general blood test indicators returned to their pre-operative (or similar) values in all participants, and regeneration of the transplant was confirmed scintigraphically in 95.3% of the cases. In 12 clinical trials, the levels of IgM returned to normal values; in 3 trials, IgM levels were higher in patients with splenic autografts than in patients after flat resections; and the other 3 trials found no difference in IgM levels between these groups. Complications of the engraftment per se were encountered in 3.7% of the cases; these included intestinal obstruction in four patients and subdiaphragmatic abscess in one patient. The incidence of delayed severe infections was evaluated in five trials, with a total of one case (pneumonia) recorded [65].

Apart from their pre-clinical significance, experimental models of splenic transplantation are of undoubted theoretical interest. The famously complex tissue architecture of the spleen is reconstituted both morphologically and functionally after the total collapse. The depth of the initial degradation can be explained by details of the technique: the surgeon creates no anastomosis between the graft vasculature and the local microcirculatory bed at the host site. Thus, the limiting stage (and also the critical stage, burdening and jeopardizing the whole process) is the ingrowth of new vessels into the autograft. Regeneration of splenic parenchyma is delayed until the vascular connection has been established. In terms of the repair process dynamics and angiogenesis, as well as hematopoietic and immune recovery at systemic level, regeneration of splenic autografts represents a unique model. Cellular sources of splenic regeneration are also of special interest. It is not clear which cells are responsible for the replacement of the destroyed splenic architecture: hematopoietic stem cells that arrive from circulation to populate the niche formed de novo by reticular cells or the resident stem cell lineages that survive within the graft despite the severity of the early necrosis. These questions have long been asked [49], but are only recently being answered [66].

The splenic stroma is known to consist of reticular tissue constituted by respective cells and fibers [66,67]. In seminal works on hematopoiesis, spleens devoid of hematopoietic cells by a lethal dose of ionizing radiation were capable of harboring proliferation and differentiation of newly transplanted hematopoietic cells [68]. These findings indicate the possibility of colonization of the autograft by circulating hematopoietic progenitors after elimination of its own hematopoietic lineages through necrosis. Such a scenario is all the more likely given that migration of hematopoietic stem cells from the bone marrow to the spleen in intact animals has been confirmed experimentally [69].

The spleen has a niche for hematopoietic stem cells, allegedly located in perivascular spaces [70]. The niche comprises endothelial cells, PDGFRb+ mesenchymal cells, and perivascular reticular cells of the red pulp which produce the stem cell factor (SCF) and CXCL12 necessary for hematopoiesis [71]. Some of the perivascular reticular cells of the red pulp also express Tcf21 and PDGFRb, thus being a unique Tcf21/PDGFRb/SCF/CXCL12 quadruple-positive cell type [72]. Stromal cells of the splenic hematopoietic niche are capable of myelopoiesis maintenance in vitro, as well as upon transplantation beneath the renal capsule in NOD/SCID mice [68,73].

Despite the existence of the authentic splenic niche that supports proliferation and differentiation of hematopoietic cells, determination of cellular sources for the splenic autograft regeneration is still an issue. One study showed that splenic capsule of 3-day-old mice transplanted allogeneically beneath the kidney capsule was able to induce genesis of a spleen with histological structure indistinguishable from the normal mouse spleen. Importantly, lymphocytes of the newly formed spleen were of the host origin [74]; the same was true in transplantations of whole embryonic spleens [75]. Stromal cells of the splenic capsule, which induced the heterotopic splenogenesis, exhibited the CD45+CD32 CD192CD11b2CD4+IL-7R+ hematopoietic lymphoid tissue inducer (LTi) phenotypes. Dynamic phenotyping of stromal cells isolated from splenic capsule in the course of postnatal development revealed a specific capacity of CD31+MAdCAM-1+LTbR+ cells with the assistance of lymphotoxin α1β2 (LT) signaling to induce the heterotopic splenogenesis [74,76]. These interesting findings are consistent with the higher regenerative capacity of splenic autografts from younger animals observed in earlier studies. The authors demonstrated that splenic capsules of >8-day-old animals failed to induce splenogenesis due to a sharp decline of CD31+MAdCAM-1+LTbR+ cells [74,76]. Nevertheless, as splenic autografts of mature animals regenerate well, the model may need a refinement.

Another potential cellular source of the splenic parenchyma in regenerating autologous splenografts consists of their own low-differentiated cells that survive during the necrotic phase. Some early findings suggested that the perivascular reticular cells may trans-differentiate into hematopoietic cells [23,33].

The evidence presented in this section suggests that the heterotopic transplantations of the spleen model is more complex for investigating splenic regeneration, but this approach may be a promising method for alleviating post-splenectomy syndrome. The spleen transplant is capable of properly clearing aging erythrocytes and bacteria and producing antibodies. Although in the contemporary perspective this is ambiguous, it is important to recognize the primacy of the hypothesis that autografts regenerate at the expense of their own cells. A more convincing concept has yet to be developed. The proper functional activity of the spleen certainly affects other organs of the abdominal cavity, and the issue of this mutual influence will be considered in the next section of our review.

**Table 2 life-12-00626-t002:** Spleen regeneration after heterotopic transplantations.

	Authors	Animal, Autograft Localization	Regeneration Period	The Effect of Autotransplantation
1.	Manley and Marine [34,35]	Rabbit, subcutaneous	80 days	
2.	Perla [36]	Rat, abdomen wall	12–21 days	
3.	Calder and Scholar [33]	Rat, mouse, omentum	30 days	
4.	Cameron and Rhee [23]	Rat, mouse, omentum	60 days	
5.	Braga et al. [37]	Rat, mesenterium	60 days	
6.	Han et al. [47]	Rat, liver lobe	35 days	
7.	Han et al. [47]	Rat, mesenterium	84 days	
8.	Miko et al. [52]	Mouse, omentum	42 days	Clearance of senescent erythrocytes from the blood, decreased platelet count and fibrinogen levels, recovery of IgM levels, a numbers of the circulating CD3+ T and CD19+ B cells remained reduced
9.	Sipka et al. [53]	Mouse, omentum		Clearance of senescent erythrocytes from the blood, decreased platelet count and fibrinogen levels
10.	Patel et al. [57]	Rat, omentum		Anti-pneumococcal defense
11.	Leemans et al. [40]	Rat, omentum		Spleen autotransplants improve humoral response to pneumococcal capsular polysaccharides
12.	Marques et al. [38,63]	Rat, omentum		Efficient clearance of *E. coli* and pneumococci
13.	Teixeira [64]	Mouse, retroperitoneum		Production of high titers of *S. aureus*-specific IgM and IgG_1_

## 4. Splenic Influence on Repair Processes in the Liver

The immune system should be vigilant. Any minor failure in its functioning immediately affects the homeostasis, as exemplified by the post-splenectomy syndrome. Meanwhile, splenectomy is absolutely indicated in certain pathological conditions of the liver. The close relationship between the two organs is reflected by the concept of hepatosplenic axis (Figure 2) [77,78]. The apparent participation of the spleen in the regulation of hepatic repair can be further redefined as the role of immunity in regeneration. Pioneering research in this field was carried out by Anna G. Babaeva and her school [79,80,81].

The spleen and the liver are anatomically connected via portal circulation and have shared responsibilities (immune, barrier, metabolic, and hematopoietic). Clinical experience shows that liver diseases often disrupt the normal splenic architecture [77,78]. At the same time, splenectomy has a positive effect on hepatic repair, though its mechanisms remain understudied [82,83].

Apart from experimental models, a significant portion of the evidence on the potential splenic involvement was obtained in patients with liver cirrhosis. The disease is accompanied by excessive production of extracellular matrix by the activated stellate cells (Ito cells) of the liver [84,85]. The activation is triggered by multiple soluble factors, most prominently TGFβ1 [85]. The elevated levels of pro-fibrotic TGFβ1, characteristic of hepatitis eventually resolved in cirrhosis, may significantly depend on the increased production of this factor by activated macrophages of the splenic red pulp. In a rat model of thioacetamide-induced cirrhosis, splenectomy leads to a decrease in blood levels of TGFβ1, considered beneficial for reparative processes in the damaged liver [86]. Consistent with these experimental findings, immunohistochemical assessment of splenic tissues in patients with cirrhosis reveals increased content of TGFβ1 and its colocalization with CD68+ cells (macrophages) [87].

The impact of splenectomy on macrophage and lymphocyte populations of the liver in cirrhosis was emphasized in a number of studies. In a mouse model of concanavalin-induced hepatitis and cirrhosis, splenectomy promotes polarization of liver macrophages towards anti-inflammatory M2 phenotypes, which support the recovery [88]. In mice with thioacetamide-induced cirrhosis, splenectomy reduces the degree of fibrotic lesions while enhancing hepatocyte proliferation and augmenting the numbers of Ly-6C(lo) monocytes and macrophages [89]. On the other hand, splenectomy in rats with induced cirrhosis alleviates the damage, in particular, through increased production of TNFα by liver macrophages; at that, the macrophage numbers remain unaltered [90]. It is noteworthy that in rats without liver damage, the liver reacts to splenectomy by enhanced proliferation of hepatocytes. Hepatic macrophages react as well: CD68+ cells increase in number, whereas the numbers of CD206+ cells decrease, with concomitant enhancement of *Il6, Il10, Tnfa, Hgf*, and *Nos2* expression in the liver [91].

The heavy-duty hepatosplenic circulation provides a permanent opportunity for the transportation of splenic monocytes/macrophages to the liver. On systemic scale, the spleen is viewed as a stock of monocytes to be released on demand for transportation and deployment at inflammatory foci. This is true for a number of murine models, including ischemic myocardial damage [92], ischemic brain damage [93], concussion spinal cord injury [94], and muscular dystrophy in mice [95]. However, the effect may as well be disease-specific; for instance, in a mouse model of lung carcinoma, the majority of monocytes arrive to the tumor directly from the bone marrow, bypassing the spleen [96]. The arrival of monocytes/macrophages to the liver in the aftermath of hepatotoxic damage or resection has been demonstrated [97,98], albeit without clear specification of their source. In our setting, intrasplenic injections of mesenchymal stromal cells (MSCs) labeled with a vital dye PKH26 led to appearance of PKH26-positive CD68+ cells (macrophages) in the liver 24 h after the injections [99]. However, whether these are liver macrophages that have engulfed the labeled MSCs arriving from the spleen, rather than splenic macrophages that have engulfed MSCs on the spot before migrating to the liver, remains unknown.

Migration of monocytes from the spleen to the liver is disputable; for splenic lymphocytes, this route has been confirmed. In a murine model of cirrhosis, the spleen becomes progressively depleted of CD4+ (helper) T lymphocytes, with a simultaneous increase in the content of Th2 lymphocytes (thought to augment fibrosis) in the liver. Under these circumstances splenectomy restores the Th1/Th2 balance and alleviates the fibrosis [100]. Apart from the lymphocytes arriving from the spleen, the liver harbors several minor lymphocyte subpopulations including γδT cells [101], NK cells [102,103], and NKT cells [104], which exert modulatory effects on liver repair [105]. The impact of splenectomy on these subpopulations remains unknown.

Splenectomy (splenectomized status) is also beneficial for regeneration of the liver after massive resections. The majority of studies emphasize the elevated rates of hepatocyte proliferation in splenectomized animals, although its mechanistic causes have to be specified. The variants include (1) resolution of portal hypertension; (2) mitigation of the damage to sinusoidal endothelium; (3) alleviation of the inflammatory side effects by inhibiting the synthesis of pro-inflammatory cytokines, as well as the rates of macrophage and neutrophil infiltration; and (4) hepatocyte apoptosis inhibition [77,106].

Splenectomy complementing 90% resections of the liver volume is accompanied by decreased expression of multiple acute phase markers in the liver remnant and increased expression of *heme oxygenase-1* gene, considered beneficial for the repair [107]. Physical removal of the spleen abrogates the inflow of pro-inflammatory cytokines that cause hepatocyte damage, as well as the proliferation blocker TGFβ1 [108,109,110]. These conditions enhance the synthesis of HMOX1 in the liver, which inhibits the activity of TNFα with a net cytoprotective effect on hepatocytes; moreover, the synthesis of TGFβ1 and its receptor TGFβ RII decreases, while the synthesis of HGF and its receptor c-met increases [108,109,111,112]. Other studies argue that the beneficial effect may be due to the withdrawal of IL10, which is a confirmed inhibitor of hepatocyte proliferation. IL10-deficient mice exhibit higher rates of hepatocyte proliferation in response to resections compared to ordinary animals. Partial hepatectomy promotes increased expression of IL10 not only in the liver, but also in the spleen; accordingly, splenectomy cuts off the inflow of IL10 via portal vein [113]. A similar positive correlation between splenectomy and liver repair is observed in liver transplantation models. The benefits include a decrease in portal hypertension and alleviation of endothelial damage, apoptosis, and pro-inflammatory cytokine synthesis [114].

On the other hand, some models question, and even disprove, the benefits of splenectomy for liver repair. For instance, Babaeva et al. observed the opposite, inhibitory effect of splenectomy on the compensatory growth of the liver after resections. The strength of the effect did not depend on the time lapse between the two interventions (splenectomy followed by liver resection); [115]. At the same time, splenectomy promoted a significant increase in the volume of intact liver through increased hepatocyte proliferation [115].

Contemporary studies on the role of immunity in regeneration involve model animals depleted of particular lymphocyte populations. The data obtained in such models are often controversial, which reflects the complexity of the regulatory mechanisms. In mice depleted of T cells, resections of 70% liver volume have lower rates of necrotic complications than in ordinary mice [116]. Similar results were obtained in a model of lipopolysaccharide-induced hepatitis; animals depleted of T or B cells revealed lower degree of hepatotoxic damage and better survival [117]. At the same time, the block of hepatocyte proliferation in rats depleted of T cells or NK cells is accompanied by the lack of proliferative response from hepatic oval cells, by contrast with the control animals, in which only the hepatocyte proliferation remained blocked [118].

Other examples of lymphoid cell participation in repair processes include experiments with adoptive transfer of lymphocytes from actively regenerating organs to the orthotopic locations in non-operated syngeneic animals [115,119]. Upon the transfer, lymphocytes retained their regeneration-supporting capacity to a degree depending on the organ, the phase of repair, and the type (population) of the lymphocytes. Transfers of helper T cells had the most pronounced effect [115,119]. The nature of regeneration-activating signals in this case is obscure; possible transmitters are microRNA molecules contained in microvesicles and exosomes secreted by lymphocytes and other cell types [120].

Thus, immune cells inside the liver, and some of those outside it, may influence repair processes within the organ or its remnant. The impact can be either activating or inhibiting; the latter is exerted by T killers, NK cells, and NKT cells. The stimulatory effect of splenectomy on the hepatic recovery after various types of damage, as well as the adoptive transfer of lymphocytes as the means for boosting liver repair, are subject to further investigation. Overall, these results indicate that the liver and spleen actively influence each other both at the level of cell migration and at the level of cytokine balance.

## 5. Conclusions

The spleen has been traditionally regarded an accessory immune organ. However, this view is challenged by severe infectious and tumorigenic consequences of splenectomy. Apart from its prominent role in immunogenesis, the spleen appears to control hepatic repair, as has been confirmed in a number of experimental models. The spleen is also a depot of monocytes, wherefrom they migrate to damaged organs. The problem of splenic regeneration is of high clinical relevance; in this regard, the most important frontier is splenic autografts. The possibility of heterotopic autologous transplantation of splenic fragments has been comprehensively assessed in experimental models and a number of clinical trials. The autografts successfully restore the normal splenic architecture of red pulp and white pulp with the newly formed periarteriolar lymphoid sheaths and lymphoid follicles. The issues still waiting to be explored include the extent of functional recovery and stability of T and B lymphopoiesis, as well as cellular sources of de novo splenogenesis in heterotopic autografts.

## Figures and Tables

**Figure 1 life-12-00626-f001:**
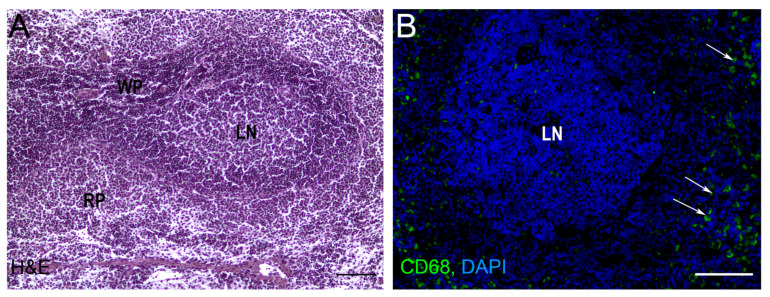
Rat spleen histology. (**A**) Light microscopy, hematoxylin-eosin staining (H&E), bars, 200 μm. (**B**) Cryosection of spleen tissue after anti-CD68 (FITC) immunostaining. The nuclei are counterstained with DAPI. RP—red pulp, WP—white pulp, LN—lymphatic nodule (follicle), bars, 200 μm. Arrows indicate CD68 + macrophages in red pulp. Original image generated in the author’s laboratory.

**Figure 2 life-12-00626-f002:**
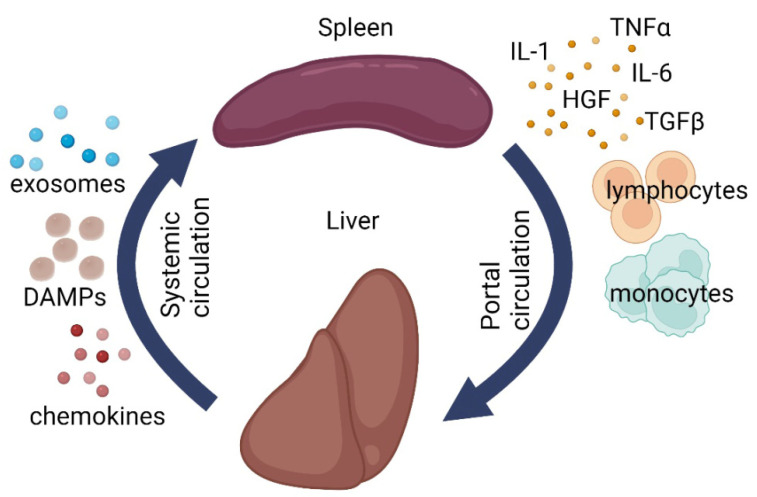
Schematic illustration of liver-splenic axis. DAMPs—damage-associated molecular patterns, IL1—interleukin 1, IL6—interleukin 6, TGF-β—transforming growth factor beta, TNFα—tumor necrosis factor alpha, HGF—hepatocyte growth factor.

## Data Availability

Not Applicable.

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
