# Peer review of "Spleen: Reparative Regeneration and Influence on Liver"

_life, 2022, doi:10.3390/life12050626_

Round 1

Reviewer 1 Report

No suggestions 

Author Response

Dear Reviewer,

Thank you for appreciating our article. In the revised version of the manuscript, a number of remarks have been corrected in accordance with the comments of other reviewers. Thank you.

Reviewer 2 Report

The article is well described, but the topic is very complex to be understood right away without a couple of pictures such as an outline, which could be very useful to better understand the text especially for "heterotopic spleen transplant" and " 'Splenic Influence in the Liver Repair Process ”chapters.

Some formatting errors

Author Response

Dear Reviewer,

Thank you for appreciating our article. In the revised version of the manuscript, we added pictures and revised several text fragments. Hope you find the changes satisfied. Thank you!

Reviewer 3 Report

The review article titled, "REPARATIVE REGENERATION OF THE SPLEEN. INFLU-ENCE OF THE SPLEEN ON REPARATIVE PROCESSES IN THE LIVER “is of importance in the field because of possible transplantation applications. However, following are the concerns to improve the quality of the manuscript:

  1. The title needs to be revised to a simpler one such as: “Role of less understood spleen in liver regeneration and immunological competence”.
  2. As the abstract says, “review considers experimental findings on splenic repair, obtained in two types of small animal models: splenic resections and autologous transplantations of splenic tissue, the sections need to be designed in the same manner. The types of animal models and the experimental findings need to be elaborated.
  3. Illustrations/figures possibly for each of the sections and graphical abstract need to be added.
  4. The sections of the manuscript sound disconnected.
  5. Specific examples from the clinical scenario need to be included.
  6. Specific examples with large animals such as dogs, horses and farm animals are also suggested.
  7. First section including the anatomy, histology of the spleen and its normal functioning need to be included.

Author Response

  1. The title needs to be revised to a simpler one such as: “Role of less understood spleen in liver regeneration and immunological competence”.

Dear Reviewer! Thank you for your careful reading and comments regarding the manuscript. We have tried to answer all points and hope you find the changes satisfied. According to your recommendation we changed the title to: “Spleen: reparative regeneration and influence on liver”

  1. As the abstract says, “review considers experimental findings on splenic repair, obtained in two types of small animal models: splenic resections and autologous transplantations of splenic tissue, the sections need to be designed in the same manner. The types of animal models and the experimental findings need to be elaborated.

Thank you for this note. We added this information in the Abstract.

  1. Illustrations/figures possibly for each of the sections and graphical abstract need to be added.

Thank you for this  comment. Graphical abstract is available in the separate section in the submitting form. With all our respect, we should note that the two main sections are accompanied by corresponding tables illustrating the material presented. According to your recommendation we have added a new illustration to the section “Splenic influence on repair processes in the liver”, thank you!

  1. The sections of the manuscript sound disconnected.

Thank you for this note. We added the connecting sentences in the text.

  1. Specific examples from the clinical scenario need to be included.

Thank you. We added the information concerning сlinical aspects in the text (page 2).

  1. Specific examples with large animals such as dogs, horses and farm animals are also suggested.

Thank you for this note. We added the recommended information in the text (page 3).

  1. First section including the anatomy, histology of the spleen and its normal functioning need to be included.

Thank you. We included the following paragraph in the Introduction section (page 2): “The spleen is  the largest lymphoid unpaired parenchymal organ of the abdominal cavity found in all vertebrates. The spleen is functionally and morphologically divided into the red and white pulp with marginal (in rodents) or perifollicular zone (in humans) between them. Blood circulation in the spleen is open: blood enters the tissues via the trabecular artery and becomes the central artery which gives rise to many branches which enter the red pulp and surround the white pulp. The red pulp is responsible for blood filtration and removes opsonized, damaged and dying cells from circulation. It also serves as the depot of the organism`s iron, as well as red blood cells, monocytes and platelets. The spleen, as the largest secondary lymphoid organ, contains about a quarter of the body's lymphocytes and initiates an immune response to blood antigens. Although adaptive immune responses to such antigens are realized in the white pulp, the cells of innate immunity (neutrophils, monocytes, dendritic cells and macrophages) could be easily found as residents of red pulp. The white pulp is divided into T- and B-zones, similar to the lymph nodes of the immune system. The T cell zone is also called the periarteriolar lymphoid sheath because it surrounds the central arterioles and is composed of lymphocytes, reticular cells and reticular fibers. B cell zones represent follicles where clonal expansion of activated B cells can take place. Marginal zone serves to monitor the circulation for antigens and pathogens and plays an important role in antigen processing and for lymphocytes releasing from the circulation and entering the white pulp. Marginal zone mainly contains macrophages and a special subset of B cells (marginal zone B cells)”.

Round 2

Reviewer 3 Report

  1. Most of the points have been addressed.
  2. Graphical abstract is good.
  3. Figure-1, Is the rat spleen histology an original image by the authors? Normally, in a review articles, no experimental images are included. In case, the mentioned image is an original image by the author, mention the image source as the original image generated in the author’s laboratory in the figure legends.
  4. Figure-2 (Liver-Splenic axis) has been wrongly mentioned in the text as Figure 1 in Page 9, Section 4
  5. In Figure-2 (Liver-Splenic axis), please expand the abbreviations in the figure legends.

  1. Most of the points have been addressed.
  2. Graphical abstract is good.
  3. Figure-1, Is the rat spleen histology an original image by the authors? Normally, in a review articles, no experimental images are included. In case, the mentioned image is an original image by the author, mention the image source as the original image generated in the author’s laboratory in the figure legends.
  4. Figure-2 (Liver-Splenic axis) has been wrongly mentioned in the text as Figure 1 in Page 9, Section 4
  5. In Figure-2 (Liver-Splenic axis), please expand the abbreviations in the figure legends.

Author Response

Dear reviewer!

Thank you for the detailed and in-depth analysis of the submitted manuscript. We have tried to answer all your questions in detail and to the point.

The article has been corrected in accordance with the comments of the reviewers. The changes are highlighted in a different color.

  1. Figure-1, Is the rat spleen histology an original image by the authors? Normally, in a review articles, no experimental images are included. In case, the mentioned image is an original image by the author, mention the image source as the original image generated in the author’s laboratory in the figure legends.

Thank you for this note. We added this information in the figure legend.

2. Figure-2 (Liver-Splenic axis) has been wrongly mentioned in the text as Figure 1 in Page 9, Section 4.

Thank you for this note. We corrected the text.

3. In Figure-2 (Liver-Splenic axis), please expand the abbreviations in the figure legends

Thank you for this note. We added this information in the figure legend.

Yours faithfully,

Dr. Andrey Elchaninov,

Dr Andrey Elchaninov, Laboratory of Regenerative Medicine, National Medical Research Center for Obstetrics, Gynecology and Perinatology Named after Academician V.I. Kulakov of Ministry of Healthcare of Russian Federation, 117997 Moscow, Russia, [email protected]